# Optimal tactics in community pension model for defined benefit pension plans

**Jun Wang, Chunli Cui, Tian Tian** *

School of Mathematics & Statistic, Changchun University of Technology, Changchun, China

* 15937292070@163.com

## Abstract

Against the backdrop of an aging population, community pension initiatives are gaining traction, permeating societal landscapes. This study delves into the equilibrium strategy within the context of a defined benefit pension plan, employing a differential game framework with a community pension model. Hence, the model entails the company's controls over investment rates in funds, juxtaposed with employees' inclination towards a greater proportion of community pension allocation in said funds. To tackle this issue, a stochastic differential game model for pensions under a community pension scheme is formulated. Leveraging the Hamilton-Jacobi-Bellman equation, we derive the Markov Perfect Nash Equilibrium solution and optimal portfolio. Through numerical simulations, we analyze the impact of varying risk aversion levels across different parameter values on equilibrium ratios, thereby offering insights into managerial risk tolerance.

## 1. Introduction

As society progresses and the challenges of an aging population loom larger, community-based aged care emerges as a prominent fixture in senior care services. The government spearheads this model, with the family at its core and the community as its foundation. More importantly, community-based senior care services offer numerous advantages such as flexibility, affordability, and extensive coverage, crucial for bolstering both societal stability and individual family welfare. Besides, extensive academic research has addressed investment portfolio and risk management in DB-type pension schemes. Notably, Haberman and Sung [1] identified contribution rate risk and solvency risk as pivotal concerns. Scholars such as Chang [2], Cairns [3], Delong et al. [4], and Xu et al. [5] have delved into optimizing defined benefit (DB) pension plan management, often employing quadratic loss function. Similarly, Josa-Fombellida and Rincón-Zapatero. [6], Hainaut and Deelstra. [7], among others, explored optimal investment dilemmas under stochastic interest rates and mean-variance criteria.

Furthermore, Josa-Fombellida and Rincón-Zapatero [8] investigated the optimal management of comprehensive dynamic pension funds, advocating for constant proportion policies. In a subsequent study, they analyzed the management of defined benefit pension plans using a non-cooperative differential game framework, addressing asymmetric strategies under varying corporate preferences [9]. Guohui et al. [10] expanded upon this work, introducing ambiguity aversion to seeking robust equilibrium strategies in worst-case scenarios. Additional literature

**Data Availability Statement:** All relevant data are within the paper.

**Funding:** This work was supported by Natural Science Foundation of Jilin Province(CN) [grant number 20200201273JC] and National Natural

Science Foundation of China (NSFC)[grant number 11871244]. The funders had no role in study design, data collection and analysis decision to publish, or preparation of the manuscript.

**Competing interests:** The authors have declared that no competing interests exist.

studies have examined robust equilibrium games, such as Pun and Wong [11], Wang et al. [12] and Wang et al. [13]. Pun and Wong's [11] study on nonzero-sum stochastic differential games between ambiguity-averse insurers, and Ricardo and Paula's [14] investigation into the impact of Poisson uncertainty on Nash equilibrium strategies and fund surplus, building upon the framework established by Josa-Fombellida and Rincón-Zapatero [9].

In the realm of community elderly care research, Yuan et al. [15] utilized incomplete game theory to develop a model decision-making framework aimed at maximizing the net returns of community elderly care projects, integrating the community pension model with social capital participation. Their findings highlighted the pivotal role of the project's target population's government dependence and the operational capability of social capital in decision-making processes. Similarly, Chen [16] underscored the significance of home-based community elderly care, identified developmental challenges, and outlined directions for community enhancement, including the establishment of robust community management mechanisms, diversification of funding sources, enhancement of management personnel training, and creation of a conducive community environment, to provide valuable insights for stakeholders.

Despite these contributions, there exists a gap in research concerning the amalgamation of defined-benefit pensions with community pensions. This study endeavors to fill this void by introducing a stochastic differential game framework for defined benefit pensions within the community pension model, thereby incorporating game theory principles. Unlike previous studies, our focus lies on the scenario of community retirement amidst underfunding, ensuring attention not only to pension plan surpluses but also to bolstering the elderly population's quality of life. Viewing the pension plan as a dynamic interaction between the company and employees, where individuals advocate for increased investment in community pensions to safeguard their future, we utilize differential game theory as a potent tool to formulate a robust DB pension plan. We will use the differential game as a powerful tool to construct a DB pension plan. In our model, the company retains control over the proportion and portfolio of the fund pool invested in financial markets, while employees wield influence over the allocation of the fund pool towards community pension. Thus, we aim to design a distribution scheme between the company and employees that ensures equitable allocation of funds in the pension fund pool, fostering mutual satisfaction.

Congruently, a comparative analysis of our findings with earlier literature is presented in Table 1.

The remaining part of this study is organized as follows: Section 2 outlines the construction of the Community Pension Plan, Section 3 provides the Markov Perfect Nash Equilibrium solution along with corresponding optimal investment strategies and fund evolution, Section 4 presents numerical simulations, and finally, Section 5 encapsulates the conclusions drawn from this study.

**Table 1. Most relevant literature.**

| literature | Community-based elderly care | Nash equilibrium | Differential game | Multi-objective optimization |
|---|---|---|---|---|
| Josa-Fombellida and Rincón-Zapatero (2018) [9] | × | ✓ | ✓ | ✓ |
| Yuan et al.(2019) [15] | ✓ | × | ✓ | × |
| Guohui et al.(2022) [10] | × | ✓ | ✓ | ✓ |
| Zhang et al.(2023) [17] | ✓ | ✓ | × | × |
| Ricardo et al.(2023) [14] | × | ✓ | ✓ | ✓ |
| Weiwei L. (2022) [18] | ✓ | × | × | × |
| This paper | ✓ | ✓ | ✓ | ✓ |

## 2. Mathematical modeling of pension plans

In this section, we denote relevant variables concerning pension fund participants. $F(t)$ represents the value of the fund assets at time $t$, $C(t)$ denotes the cumulative pension amount determined by the company at retirement time $t$ signifies the defined benefits at time $t$, $P(t)$ represents the normal cost of contribution for all employees, $NC(t)$, $AL(t)$ denotes the actuarial liability, and $UAL$ stands for the unfunded actuarial liability, calculated as the difference between $AL(t)$ and $F(t)$. Additionally, $SC$ indicates the supplementary contribution rate at time $t$, computed as the difference between $C(t)$ and $NC(t)$.

Notably, $P(t)$, $AL(t)$, and $NC(t)$ remain constant, assuming a stable number of pensioners is stable over time without accounting for wage increases, consistent with the framework established by Haberman and Sung [1]. When the fund asset $F(t)$ matches the actuarial liability $AL(t)$ and uncertainties are absent in the plan, the normal cost $NC(t)$ represents the deterministic function of the contribution rate's value.

This adopts a spread method of fund amortization, akin to the approaches of Haberman and Sung [1] and Josa-Fombellida and Rincón-Zapatero [8], operating under the premise of a positive correlation between the unfunded actuarial liability and the supplemental contribution rate, as articulated in the following equation:

$$C(t) = NC(t) + k(AL(t) - F(t)), \tag{1}$$

where $k$ the amortization rate is a crucial factor. Assuming that the valuation of the plan is done at a constant rate (which can be determined by the regulator) $\rho$, Bowers et al. [19], linked the main components of the plan via the following equation:

$$\rho AL + NC - P = 0. \tag{2}$$

Let's consider a probability space $(\Omega, \mathcal{F}, \mathrm{P})$, where $P$ is a probability measure on $\Omega$ and $\mathcal{F} = \{\mathcal{F}_t\}_{t \geq 0}$ is generated by an n-dimensional standard Brownian motion $w = (w_1, \cdots, w_n)^\top$, that is, $\mathcal{F}_t = \sigma\{w_1(s), \cdots, w_n(s); 0 \leq s \leq t\}$. Investors have the option to trade between a risk-free bond $S_0$ and n risky assets $S_1, \cdots, S_n$, each following a geometric Brownian motion. Thus, the dynamic equation is presented as follows:

$$dS_0(t) = rS_0(t)dt, \ S_0(0) = 1, \tag{3}$$

$$dS_i(t) = S_i(t)\left(b_i dt + \sum_{j=1}^{n} \sigma_{ij} dw_j(t)\right), S_i(0) = s_i, i = 1, \cdots, n. \tag{4}$$

Furthermore, the constant $r > 0$ represents the short risk-free interest rate, $b_i > 0$ signifies the average rate of return of the risky asset $S_i$, and $\sigma_{ij} > 0$ denotes the volatility coefficients. We, therefore, assume that $b_i > r$, for each $i = 1, \cdots, n$, effectively eliminates arbitrage risk and provides the company with a heightened incentive to invest in risky assets. The matrix $\sigma_{ij}$ is denoted by $\sigma$; then, the Sharpe ratio of the combination $\theta$ is $\sigma^{-1}(b - r\bar{1})$, where $b = (b_1, \cdots, b_n)^\top$ and $\bar{1}$ is a symmetric positive definite matrix $\Sigma = \sigma\sigma^\top$, ensuring the completeness of the financial market.

Within the pension model, the company dynamically constructs an investment portfolio comprising bonds and risky assets to invest in the $u_1$ portion of the fund, while employees determine the allocation of the fund to the community pension, denoted as $u_2$, $u_1, u_2 \in \mathbb{R}$, $0 < u_1, u_2 < 1$, $0 < u_1 + u_2 < 1$. Let $\lambda_i$ be the share of the investment in risky assets $S_i$, $i \in \{1, \cdots, n\}$, and $u_1 F - \sum_{i=1}^{n} \lambda_i$ be the share of the investment in the fund. If $\Lambda = (\lambda_1, \cdots, \lambda_n)^\top$, then the variation

function of the fund follows the dynamics

$$dF = \sum_{i=1}^{n} \lambda_i(t) \frac{dS_i(t)}{S_i(t)} + \left( u_1 F(t) - \sum_{i=1}^{n} \lambda_i \right) \frac{dS_0(t)}{S_0(t)} + (C(t) - P(t) - u_2 F(t))dt, \tag{5}$$

after substituting (1), (3), and (4) in (5), as represented in the following equation.

$$dF = ((ru_1 - u_2 - k)F(t) + NC(t) + kAL(t) - P(t) + \Lambda^{\top}(b - r\bar{1}))dt + \Lambda^{\top}\sigma dw(t). \tag{6}$$

Assuming the contribution rate equals the fixed valuation rate of the liability, denoted by $k = \rho$, then Eq (6) becomes:

$$dF = ((ru_1 - u_2 - \rho)F(t) + \Lambda^{\top}(b - r\bar{1}))dt + \Lambda^{\top}\sigma dw(t), \tag{7}$$

with the initial condition $F(0) = F_0 > 0$.

Portfolio $\Lambda(t)$ is a testable procedure for $\mathbb{R}^n$ to $\{\mathcal{F}_t\}$; and for $\forall t < \infty$, we have the following formulation:

$$\int_0^{\infty} \|\Lambda(s)\|^2 ds < \infty \text{ a.s.}$$

Let's define the class of admissible strategies of the company as A, $\Lambda \in A$. We aim to attain satisfactory outcomes for both sides of the strategy. Hence, the company's objective function is expressed as follows:

$$J_1(F; u_1, u_2, \Lambda) = \mathrm{E}_F \int_0^{\infty} e^{-\alpha t} m(u_1(t))dt, \tag{8}$$

where $m$ is the utility function representing corporate fund evolution and $\alpha > 0$ is the company's time preference. Following the same principle, we formulate the objective function for employees as:

$$J_2(F; u_1, u_2, \Lambda) = \mathrm{E}_F \int_0^{\infty} e^{-\beta t} n(u_2(t))dt, \tag{9}$$

where $n$ signifies the utility function for employees' community pension and $\beta > 0$ represents the employees' time preference.

To derive an explicit solution, we adopt the strictly concave CRRA utility function.

$$m(u_1(t)) = \frac{(u_1 F)^{1-\gamma} - 1}{1 - \gamma}, \gamma > 0,$$

$$n(u_2(t)) = \frac{(u_2 F)^{1-\delta} - 1}{1 - \delta}, \delta > 0.$$

## 3. Nash equilibrium strategy

In the above model, the players consist of enterprises and individuals. Remarkably, the pension model operates as a two-person non-cooperative differential game, where the relevant solution concept is the Markov perfect Nash equilibrium. Likewise, the Nash equilibrium solution

embodies a set of strategies acceptable to all players in the game, representing a scenario where no unilateral decision change by any party could lead to a better outcome than the one achieved at Nash equilibrium.

**Definition 3.1** Consider a probability space $(\Omega, \mathcal{F}, P)$, where $P$ is a probability measure on $\Omega$, and $\mathcal{F} = \{\mathcal{F}_t\}_{t \geq 0}$ is generated by n-dimensional standard Brownian motion $w = (w_1, \cdots, w_n)^\top$, that is, $\mathcal{F}_t = \sigma\{w_1(s), \cdots, w_n(s); 0 \leq s \leq t\}$. We define the investment portfolio $\Lambda(t)$ as a testable procedure for $\mathbb{R}^n$ to $\{\mathcal{F}_t\}$, for $\forall t < \infty$, $\Lambda$ satisfy integrability conditions, as depicted in equation below:

$$\int_0^\infty \|\Lambda(s)\|^2 ds < \infty \text{ a.s.}$$

Assuming the Markov perfect Nash equilibrium solution of the defined benefit pension game with a community pension is $(u_1^*, u_2^*) \in \mathbb{R} \times \mathbb{R}$ and the best investment plan is $\Lambda^* \in A$. Therefore, for any $F > 0$, $(u_1, u_2) \in \mathbb{R}$, we have:

$$J_1(F; u_1^*, u_2^*, \Lambda^*) \geq J_1(F; u_1, u_2^*, \Lambda),$$

$$J_2(F; u_1^*, u_2^*, \Lambda^*) \geq J_2(F; u_1^*, u_2, \Lambda^*).$$

Thus, we employ the value function method to obtain the Nash equilibrium solution and introduce Theorem 3.1 thereafter.

**Theorem 3.1** It states that the stochastic formulas for a quantitative differential game of specified duration contain a stochastic differential equation describing the evolution of the state and cost functionals, as expressed below:

$$dx_t = F(t, x_t, u^1(t), \ldots, u^N(t))dt + \tilde{\sigma}(t, x_t)dw_t, x_t|_{t=0} = x_0$$

Where $x_t$ is the game trajectory, $u^i$ is the decision, and $P^i$ is the i$^{\text{th}}$ player, which describes the evolution of the state, and N contains the cost functionals. $L^i$ is the cost function of $P^i$ in a differential game of fixed duration.

Going further, we describe the evolution of the N cost functions of the state, as follows:

$$L^i(u^1, \ldots, u^N) = \int_0^T g^i(t, x_1, u^1(t), \ldots, u^N(t))dt + q^i(x_T); i \in N$$

If $\Gamma^i$ ($i \in N$) represents $P^i$'s strategy space (i.e. compatible with the deterministic information pattern of the game), $\gamma^i$ is $P^i$'s allowed strategy. It is therefore assumed that $F$ and $\sigma$ meet the requirements of Thm5.2[18] and that $\sigma$ is not singular.

For an N-person nonzero-sum stochastic differential game of prescribed fixed duration [0, $T$], an N-tuple of feedback strategies $\{\gamma^{i^*} \in \Gamma^i; i \in N\}$ provides a Nash equilibrium solution if certain smooth functions $W^i:[0,T] \times R^n \to R, i \in N$ satisfy coupled semi-linear parabolic partial differential equations. As stated below:

$$-\frac{\partial W^i}{\partial t} - \frac{1}{2}\sum_{k,j}\sigma^{kj}(t, x)\frac{\partial^2 W^i}{\partial x_k \partial x_j}$$

$$= \min_{u^i \in S^i}[\nabla_x W^i \cdot \tilde{F}^{i^*}(t, x, u^i) + \tilde{g}^{i^*}(t, x, u^i)]$$

$$= \nabla_x W^i \cdot \tilde{F}^{i^*}(t, x, \gamma^{i^*}(t, x)) + \tilde{g}^{i^*}(t, x);$$

$$W^i(T, x) = q^i(x) \quad (i \in N),$$

where $\sigma^{kj}$ is the $kj^{\text{th}}$ element of the symmetric matrix $\tilde{\sigma}\tilde{\sigma}'$,

$$\tilde{F}^{i*}(t, x, u^i) \triangleq F(t, x, \{\gamma^*_{-i}, u^i\}),$$

$$\tilde{g}^{i*}(t, x, u^i) \triangleq g^i(t, x, \{\gamma^*_{-i}, u^i\}),$$

$$\{\gamma^*_{-i}, u^i\} \triangleq (\gamma^{1*}(t, x), \dots, \gamma^{i-1*}(t, x), u^i, \gamma^{i+1*}(t, x), \dots, \gamma^{N*}(t, x))$$

Given the existence theorem of the Nash equilibrium solution in our model is clearly stated, we assume that the value functions of this model are $V_1$ and $V_2$. As mathematically represented below:

$$V_1(F) = J_1(F; u_1^*, u_2^*, \Lambda^*),$$

$$V_2(F) = J_2(F; u_1^*, u_2^*, \Lambda^*).$$

Consequently, we came up with the following proposition.

**Proposition 3.1** This proposition states that given the specific parameter settings in the game, where the constant A is negative and the constant B is positive, both finite, the value function of the company and its employees in the Nash pension game (7), (8), (9) becomes:

$$V_1(F) = A \frac{F^{1-\gamma}}{1-\gamma} - \frac{1}{\alpha(1-\gamma)},$$

$$V_2(F) = B \frac{F^{1-\delta}}{1-\delta} - \frac{1}{\beta(1-\delta)},$$

respectively, where:

$$A = \left( \left( \frac{\rho + \beta}{1-\delta} + \frac{\alpha\delta}{(1-\gamma)(1-\delta)} - \frac{(1-\delta)(3\gamma - \delta) - \gamma}{2\gamma^2(1-\delta)} \theta^\top \theta \right) \frac{(1-\delta)(1-\gamma)}{\gamma + \delta - 1} (-r)^{-\frac{1-\gamma}{\gamma}} \right)^{-\gamma}, \quad (10)$$

$$B = \left( \frac{(\rho + \alpha)(1-\delta) + \beta\gamma}{\gamma + \delta - 1} + \frac{\gamma(3\delta - 1) - (\delta - 1)^2}{2\gamma(\gamma + \delta - 1)} \theta^\top \theta \right)^{-\delta}, \quad (11)$$

and satisfy:

$$-2(\beta + \rho)\gamma^3 + \gamma\delta(2\alpha\gamma + \theta^\top\theta(2 - 3\gamma + \delta)) + 2\gamma^2(\beta + \rho + \theta^\top\theta) + \theta^\top\theta(-\delta^2 - 2\gamma + \delta) < 0,$$

$$(2\alpha + 2\rho - \theta^\top\theta)\gamma - (2\alpha + 2\rho - 3\theta^\top\theta)\gamma\delta + 2\beta\gamma^2 - \theta^\top\theta(\delta - 1)^2 > 0.$$

Consequently, it represents the Markov perfect Nash equilibrium solution $(u_1^*, u_2^*)$ and the corresponding optimal investment strategy $\Lambda^*$, which is:

$$u_1^* = (-rA)^{-\frac{1}{\gamma}}, \quad (12)$$

$$u_2^* = B^{-\frac{1}{\delta}}, \quad (13)$$

$$\Lambda^* = \sum^{-1}(b - r\bar{1})\frac{1}{\gamma}F. \quad (14)$$

The parameters in (12) and (13) should satisfy $0 < u_1^* < 1$, $0 < u_2^* < 1$, and $u_1^* + u_2^* < 1$. Whereas, the equilibrium fund surplus follows a geometric Brownian motion process:

$$dF^*(t) = \left( (-r)^{\frac{\gamma-1}{\gamma}} A^{-\frac{1}{\gamma}} - B^{-\frac{1}{\delta}} - \rho - \frac{1}{\gamma} \theta^\top \theta \right) F^* dt + \frac{1}{\gamma} \theta^\top F^* dw(t). \tag{15}$$

**Proof:** To prove this, we start with the Hamilton-Jacobi-Bellman system of the Nash pension game:

$$\alpha V_1 = \max_{u_1} \left\{ \frac{(u_1 F)^{1-\gamma}}{1-\gamma} + \left[ (ru_1 - u_2 - \rho)F(t) + \Lambda^\top (b - r\bar{1}) \right] V'_1(F) + \frac{1}{2} \Lambda^\top \sum \Lambda V''_1(F) \right\},$$

$$\alpha V_1 = \max_{\Lambda} \left\{ \frac{(u_1 F)^{1-\gamma}}{1-\gamma} + \left[ (ru_1 - u_2 - \rho)F(t) + \Lambda^\top (b - r\bar{1}) \right] V'_1(F) + \frac{1}{2} \Lambda^\top \sum \Lambda V''_1(F) \right\},$$

$$\beta V_2 = \max_{u_2} \left\{ \frac{(u_2 F)^{1-\delta}}{1-\delta} + \left[ (ru_1 - u_2 - \rho)F(t) + \Lambda^\top (b - r\bar{1}) \right] V'_2(F) + \frac{1}{2} \Lambda^\top \sum \Lambda V''_2(F) \right\}.$$

Utilizing the first-order optimality conditions, we derive expressions for:

$$0 = u_1^{-\gamma} F^{1-\gamma} + rFV'_1(F) \Rightarrow u_1 = (-r)^{-\frac{1}{\gamma}} (V'_1(F))^{-\frac{1}{\gamma}} F^{-1}, \tag{16}$$

$$0 = u_2^{-\delta} F^{1-\delta} - FV'_2(F) \Rightarrow u_2 = (V'_2(F))^{-\frac{1}{\delta}} F^{-1}, \tag{17}$$

$$0 = (b - r\bar{1}) V'_1(F) + \Lambda^\top \sum V''_1(F) \Rightarrow \Lambda = -\sum^{-1} (b - r\bar{1}) \frac{V'_1(F)}{V''_1(F)}. \tag{18}$$

Substituting these expressions into the HJB system, after some simplifications, yields:

$$\begin{cases} \alpha V_1 = \frac{\gamma}{1-\gamma} (-r)^{\frac{\gamma-1}{\gamma}} (V'_1(F))^{\frac{\gamma-1}{\gamma}} - \frac{1}{1-\gamma} - (V'_2(F))^{-\frac{1}{\delta}} V'_1(F) - \rho F V'_1(F) - \frac{1}{2} \theta^\top \theta \frac{(V'_1(F))^2}{V''_1(F)} \\ \\ \beta V_2 = \frac{\delta}{1-\delta} (V'_2(F))^{\frac{\delta-1}{\delta}} - \frac{1}{1-\delta} - (-r)^{\frac{\gamma-1}{\gamma}} (V'_1(F))^{-\frac{1}{\gamma}} V'_2(F) - \rho F V'_2(F) \\ \quad\quad - \theta^\top \theta \frac{V'_1(F) V'_2(F)}{V''_1(F)} + \frac{1}{2} \theta^\top \theta \left( \frac{V'_1(F)}{V''_1(F)} \right)^2 V''_2(F) \end{cases} \tag{19}$$

Thereafter, we simplified solutions $V_1(F) = A \frac{F^{1-\gamma}}{1-\gamma} - \frac{1}{\alpha(1-\gamma)}$ and $V_2(F) = B \frac{F^{1-\delta}}{1-\delta} - \frac{1}{\beta(1-\delta)}$, where $A < 0$, $B > 0$ are suitable constants that are determined via Eq (19). Substituting the guess solution into (19), we derive equations for $F^{1-\gamma}$ and $F^{1-\delta}$, which can be obtained from the corresponding equality.

$$\frac{\alpha A}{1-\gamma} = \frac{\gamma}{1-\gamma} (-r)^{\frac{\gamma-1}{\gamma}} A^{\frac{\gamma-1}{\gamma}} - B^{-\frac{1}{\delta}} A - \rho A + \frac{1}{2} \theta^\top \theta \frac{A}{\gamma},$$

$$\frac{\beta B}{1-\delta} = \frac{\delta}{1-\delta} B^{\frac{\delta-1}{\delta}} - (-r)^{\frac{\gamma-1}{\gamma}} A^{-\frac{1}{\gamma}} B - \rho B + \theta^\top \theta \frac{B}{\gamma} - \frac{1}{2} \theta^\top \theta \frac{B\delta}{\gamma^2}.$$

Solving for A and B yields (10) and (11) from the two equations above. More so, by substituting the exact $V_1(F)$ and $V_2(F)$ into Eqs (16)–(18), we derive the Markov perfect Nash

equilibrium $(u_1^*, u_2^*)$ and the optimal investment strategy $\Lambda^*$. By incorporating $(u_1^*, u_2^*)$ and $\Lambda^*$ into the stochastic differential in Eq (7), where the evolution process of the fund is obtained by solving this stochastic differential equation. Also, we ascertain the Markov perfect Nash equilibrium and the optimal investment strategy. Equally, we obtain the optimal evolution process of the fund in Eq (15), where the solution of Eq (15) becomes:

$$F^*(t) = F \exp\left( \phi(\gamma, \delta)t + \frac{1}{\gamma}\theta^\top \int_0^t w(s)ds \right),$$

where $\phi(\gamma, \delta) = (-r)^{\frac{\gamma-1}{\gamma}}A^{-\frac{1}{\gamma}} - B^{-\frac{1}{\delta}} - \rho - \frac{1}{\gamma}\theta^\top\theta$.

Finally, we demonstrate the fulfillment of the transversality condition, ensuring the validity of the proposition. Thus:

$$\lim_{t\to\infty} e^{-\alpha t}\mathrm{E}_F V_1(F^*(t)) = \lim_{t\to\infty} e^{-\beta t}\mathrm{E}_F V_2(F^*(t)) = 0. \tag{20}$$

For any real number $a$, we observe that:

$$\mathrm{E}_F(F^*(t))^a = F^a \exp\left\{ a\left( (-r)^{\frac{\gamma-1}{\gamma}}A^{-\frac{1}{\gamma}} - B^{-\frac{1}{\delta}} - \rho - \frac{1}{\gamma}\theta^\top\theta \right)t + \frac{a^2}{2\gamma^2}\theta^\top\theta t \right\}.$$

After substituting $a$ with $1-\gamma$ and $1-\delta$, we find that Eq (20) if and only if

$$(1-\gamma)\left( (-r)^{\frac{\gamma-1}{\gamma}}A^{-\frac{1}{\gamma}} - B^{-\frac{1}{\delta}} - \rho + \frac{1-3\gamma}{2\gamma^2}\theta^\top\theta \right) < \alpha,$$

$$(1-\delta)\left( (-r)^{\frac{\gamma-1}{\gamma}}A^{-\frac{1}{\gamma}} - B^{-\frac{1}{\delta}} - \rho + \frac{1-\delta-2\gamma}{2\gamma^2}\theta^\top\theta \right) < \beta.$$

This condition holds if and only if $A<0$ and $B>0$, respectively, aligns with the assumption in the proposition.

Remark 3.1 $u_1^*, u_2^*$ in proposition 3.1, is influenced by various factors such as $\gamma$, and $\delta$. For instance, $\Lambda^*$ is a linear function of fund assets impacted by $\gamma$. As $\gamma$ increases, indicating greater risk aversion by the company, the optimal investment strategy $\Lambda^*$ decreases, as the company tends to shy away from riskier investments. Consequently, the surplus $u_1^*$ will increase. To ensure the pension plan's sound operation, the company opts for more capital allocation towards risk-free investment. Meanwhile, $u_2^*$ will decrease, reflecting a diminishing proportion of funds allocated to the community pension as more funds are invested elsewhere.

## 4. Numerical simulation

In the subsequent section on numerical simulation, we analyze the impact of different risk aversion coefficients and parameter values on the equilibrium ratios $u_1^*$ and $u_2^*$. Both bureau members are subjected to the same discount factor of 0.98, i.e. $\alpha = \beta = 0.02$. To simplify the exposition, we assume the employees' risk aversion is lower than that of the company, a reasonable assumption considering the company bears all investment risks. Thus, we take $\gamma = \delta +1$, where $\delta \in [1,9]$ and $\gamma \in [2,10]$ represent the respective risk aversion coefficients, to demonstrate varying risk attitudes in the game. In this context, $r \in [0.89\theta^\top\theta, \theta^\top\theta]$ signifies the difference in risk aversion levels. Given these assumptions, with a risk-free rate denoted as $r = 0.04$, we maintain a constant rate equal to the risk-free rate. $b = 0.06$ represents the coefficient of

return on risky assets, while $\sigma = 0.1$ indicates its volatility change. With these considerations in mind, we proceed to present our findings.

Fig 1 illustrates the correlation between $u_1^*$, the proportion of investment rate by the company and $\gamma$. It shows that as $\gamma$, so does the proportion of investment rate $u_1^*$. This relationship experiences a rapid ascent [2,6] from 0.625 to 0.9028. However, when $\gamma > 6$, further increases $u_1^*$, the rate of increase in the proportion of investment slows down and tends to stabilize around 0.95. This slowdown can be attributed to the high-risk-averse nature of the company, leading to a reduction in investment in risky assets. Consequently, the rise of $u_1^*$ implies that the company seeks to allocate more funds to risk-free assets to ensure the healthy and sustainable development of pension plans.

Fig 2 depicts the variation of $u_2^*$ influenced by $\gamma$. In addition, the value of $u_2^*$ is 0.04 when the risk aversion of both sides is small ($\delta = 1$, $\gamma = 2$). While, $u_2^*$ is more sensitive to the changes in $\gamma$ when it is small ($\gamma \in [2,4]$). This heightened sensitivity arises because an increase in the company's risk aversion prompts greater market investment, resulting in a notable decrease in the proportion of funds allocated to community pensions. As gradually increases, $u_2^*$ sharply declines to 0.0062, stabilizing thereafter at 0.005.

When market conditions are sufficiently favorable and the risk aversion of firms and employees converges, at which point we take $\gamma = \delta + v$, where $v$ approaches zero, i.e., is infinitesimally small yet greater than zero, thus, we assume $\delta \in [1,9]$, i.e., $\gamma \in [1+v, 9+v]$ always remains greater than zero. This scenario enables us to denote differing risk attitudes within the bureau. In this context, with $r \in [0.89\theta^{\mathrm{T}}\theta, \theta^T\theta]$, we derive the following outcomes.

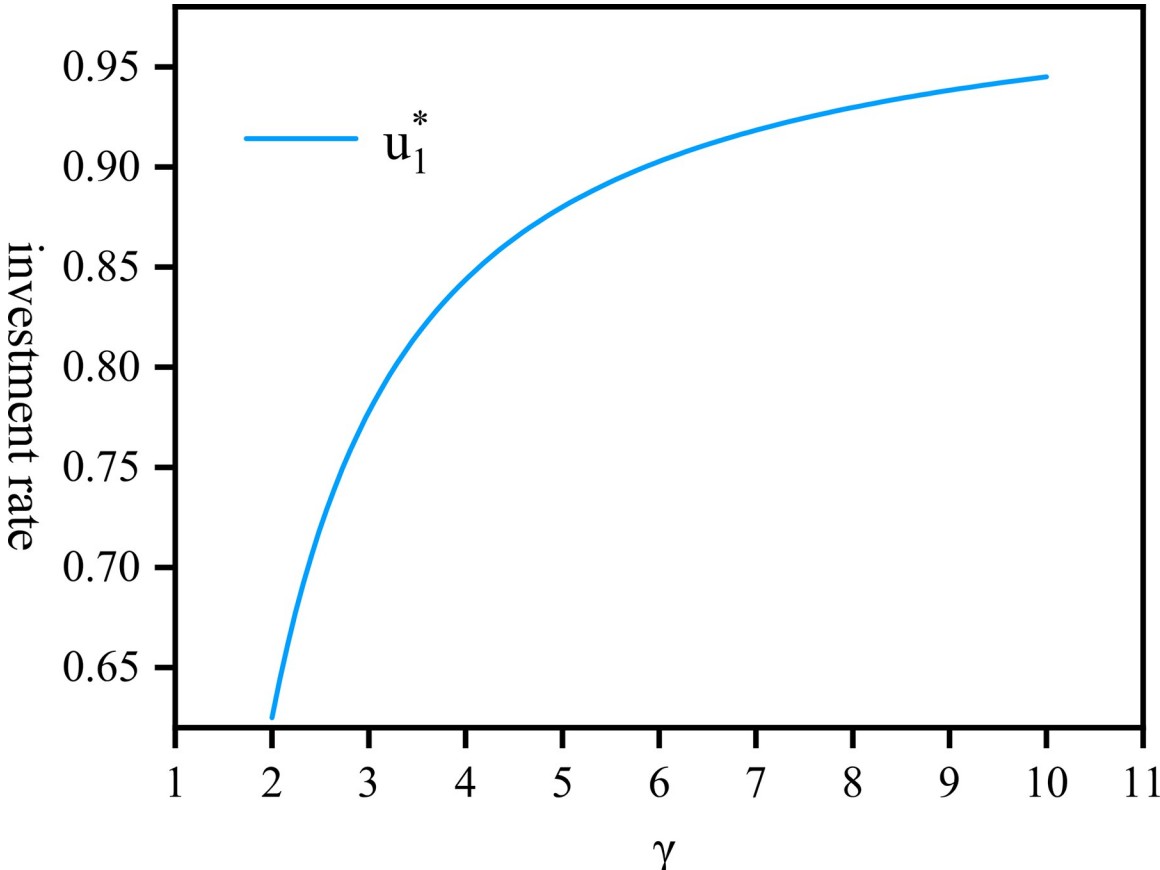

**Fig 1. The proportion of investment rate by the company draws against $\gamma \in [2,10]$.**

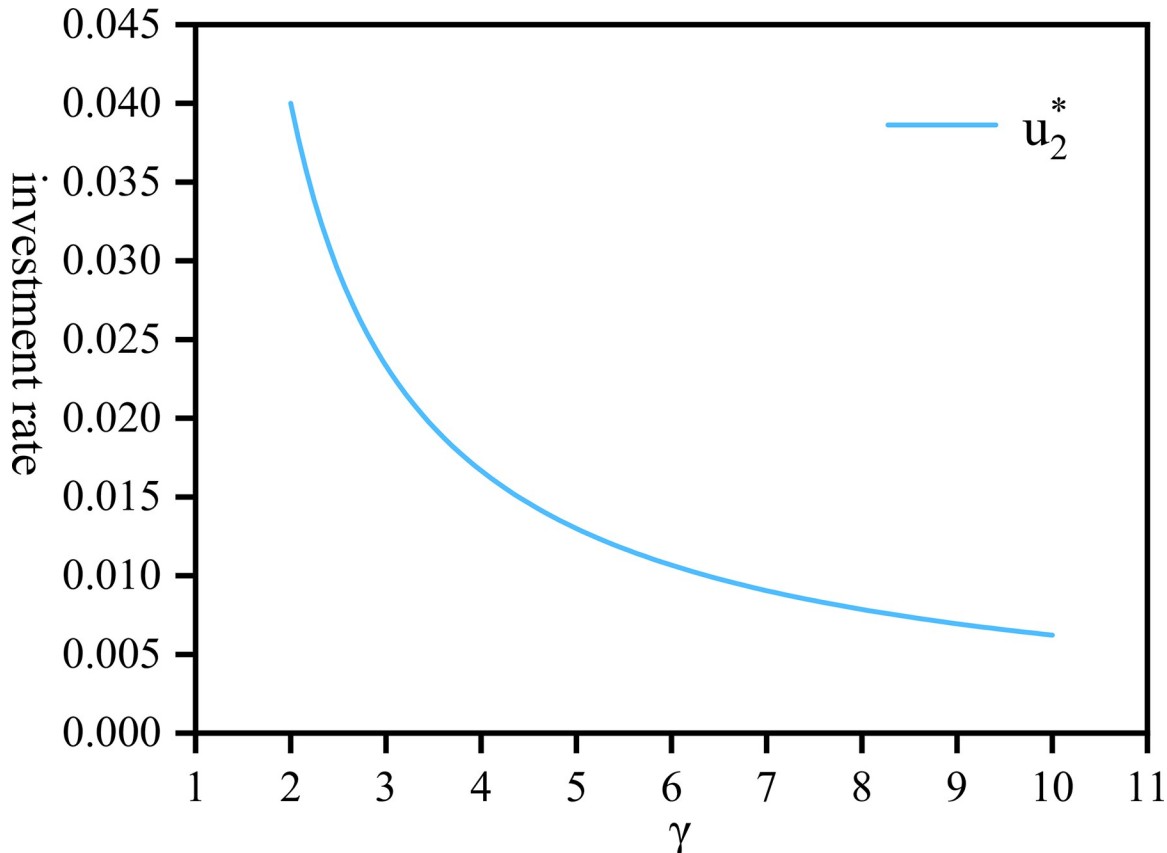

**Fig 2. The proportion of community pension by company draw against $\gamma \in [2,10]$.**

Fig 3 depicts the relationship between $u_1^*$ and $\gamma$. As $\gamma$ increases, $u_1^*$ gradually rises as well. Initially, this increase is rapid, particularly in the range of [1,4]. However, as $\gamma > 4$, surpasses a certain threshold, the upward trend of $u_1^*$ gradually flattens out and converges towards 0.9. This phenomenon occurs because highly risk-averse companies minimize their investment in risky assets, and $u_1^*$ opting instead to allocate more funds to risk-free assets to ensure the longevity and stability of the pension scheme. Fig 4 illustrates the variation of $u_2^*$ influenced by $\gamma$. When the risk aversion values of both parties are small (i.e., $\delta = 1$, $\gamma = 1$), $u_2^*$ remains at 0.06. Compared with Fig 2, and similar to Fig 3, both graphs share the same trend. However, during this period, both the company and employees are more sensitive to changes in risk factor and are more susceptible to the impact of the market economy.

For a fund pool, where in addition to investing in assets, there are community pension expenses and personal pensions to employees, hence the proportion of personal pensions is $1 - u_1^* - u_2^*$. Fig 5 represents the sum of $u_1^*$ and $u_2^*$, reflecting the company's comprehensive investment in assets and community pension, respectively, as the company's risk aversion coefficient varies. The chart indicates that as the risk aversion coefficient gradually increases, the total proportion allocated to investment and community pension also rises, eventually reaching around 95%. Conversely, Fig 6 illustrates the relationship between the personal pension function $P(t)$ and the risk aversion coefficient. When the company's risk aversion coefficient is low, the proportion allocated to personal pensions is higher. Nevertheless, as the coefficient of risk aversion increases, the personal pension proportion will gradually decrease and stabilize but will not approach zero.

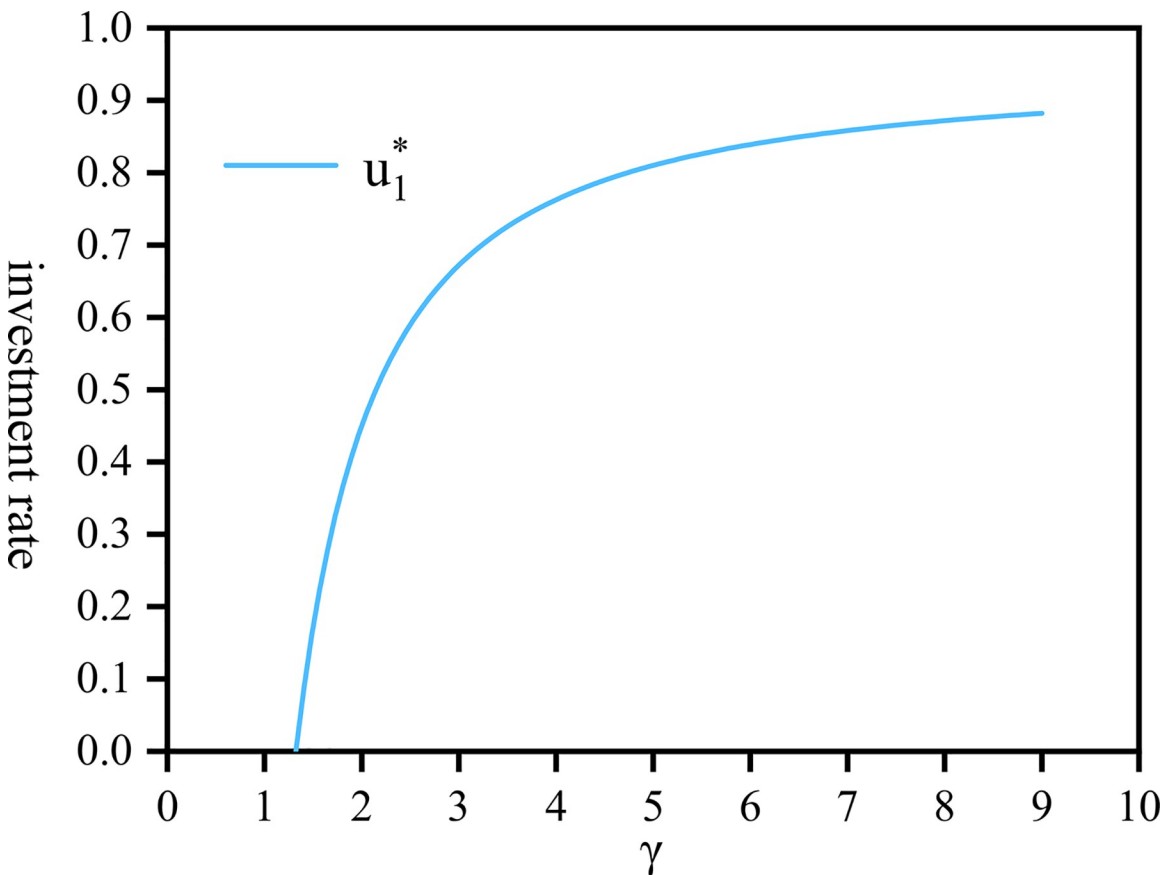

**Fig 3. The proportion of investment rate by the company draws against $\gamma \in [1+v, 9+v]$.**

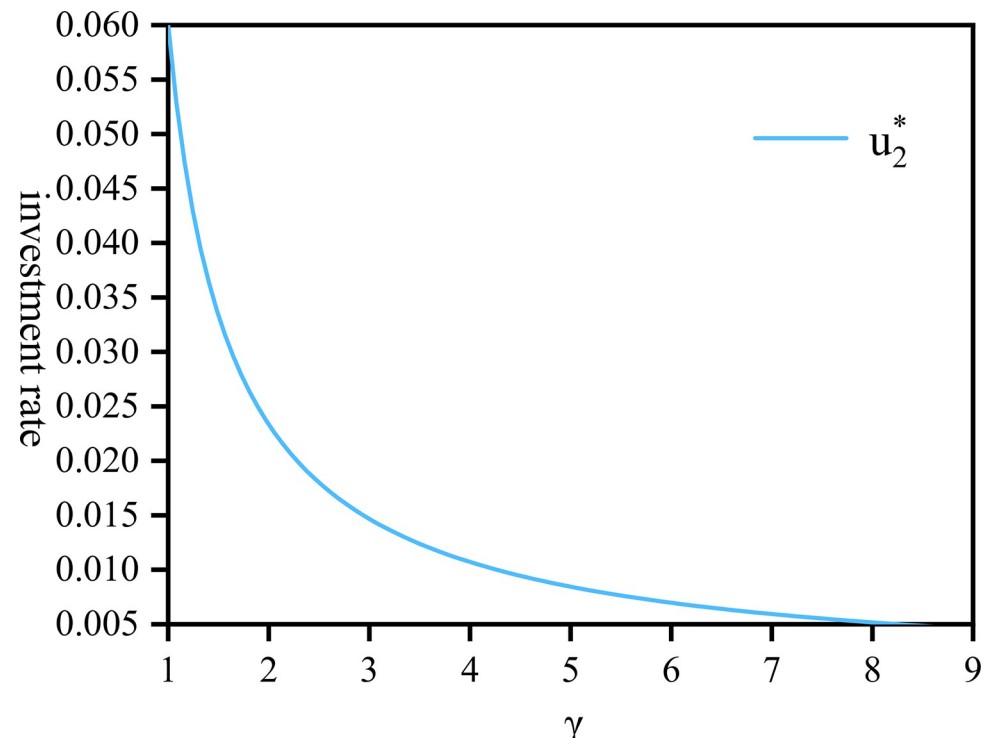

**Fig 4. The proportion of community pension by company drawn against $\gamma \in [1+v, 9+v]$.**

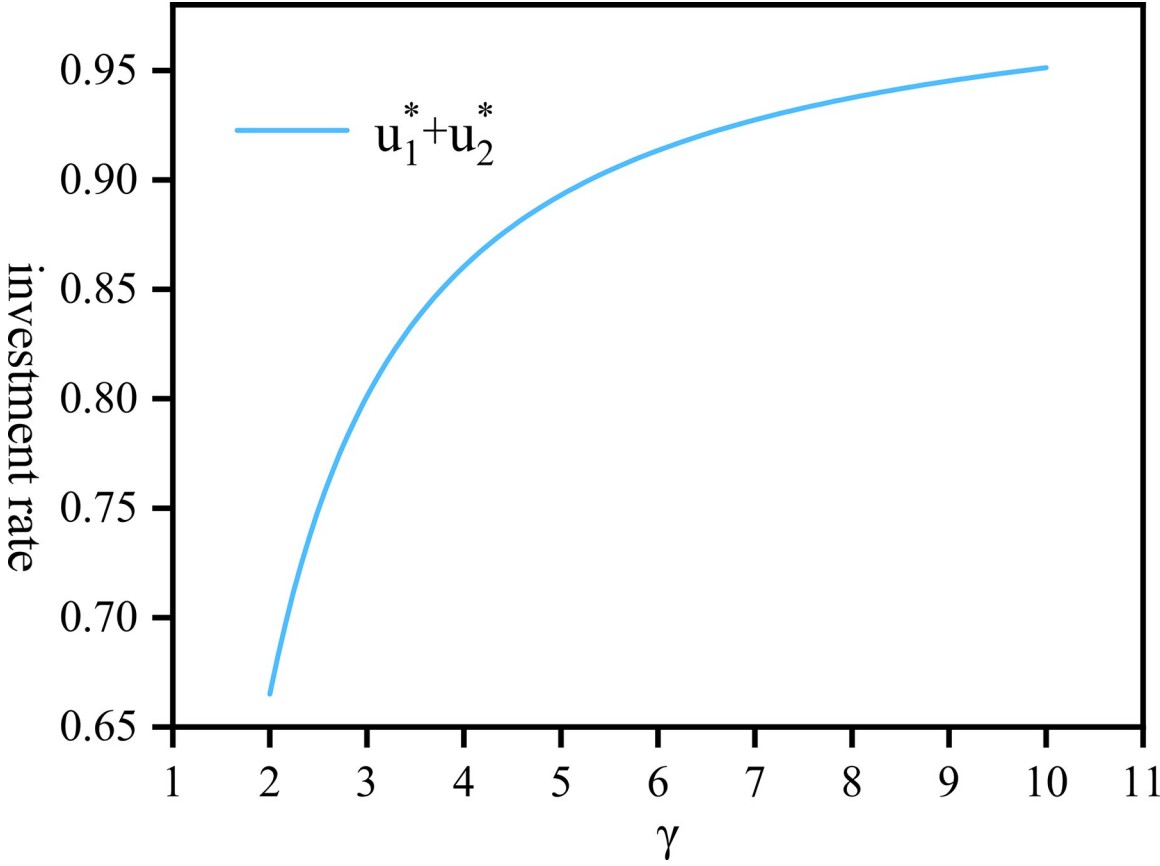

**Fig 5. The total proportion $u_1^* + u_2^*$ draw against $\gamma \in [2,10]$.**

Fig 7 presents a comparison of three scale coefficients: the investment rate, community pension ratio, and personal pension ratio. Upon comparing $u_1^*$, $u_2^*$, and $P(t)$, it becomes evident that the value of $u_1^*$ surpasses that of both $u_2^*$ and $P(t)$. This discrepancy arises because the company, acting as a rational player, strategically allocates a larger portion of fund assets toward investment when the rate of return on risky assets and the risk-free interest rate is low. This strategic investment aims to maximize financial returns, ensuring the smooth disbursement of personal pensions and the capacity to fund community pensions.

In the preceding discussion, we utilized the CRRA utility function. To establish $V_1$ and $V_2$, it is imperative that $F > \max\left\{\frac{1}{u_1}, \frac{1}{u_2}\right\}$. With the introduction of predetermined parameters, $F > 161.3$ can be determined, a condition easily satisfied (in this case, the unit of measurement of $F$ is yuan). Assuming a pension fund of $F = 200000 > 161.3$, and a pension period of 50 years post-retirement, the value function is therefore represented as follows.

$$V_1 = \max_{u_1} E_F \int_0^{50} e^{-0.02t} \frac{(u_1 F)^{1-\gamma} - 1}{1 - \gamma} dt = 50\left(1 - \frac{1}{e}\right)\left(\frac{(2\gamma^2 - \gamma - 1)^{1-\gamma} F^{1-\gamma}}{(1 - \gamma)(2\gamma^2)^{1-\gamma}} - \frac{1}{1 - \gamma}\right),$$

$$V_2 = \max_{u_2} E_F \int_0^{50} e^{-0.02t} \frac{(u_2 F)^{2-\gamma} - 1}{2 - \gamma} dt = 50\left(1 - \frac{1}{e}\right)\left(\frac{(0.06\gamma - 0.04)^{2-\gamma} F^{2-\gamma}}{(2 - \gamma)(\gamma^2 - \gamma)^{2-\gamma}} - \frac{1}{2 - \gamma}\right).$$

Value function

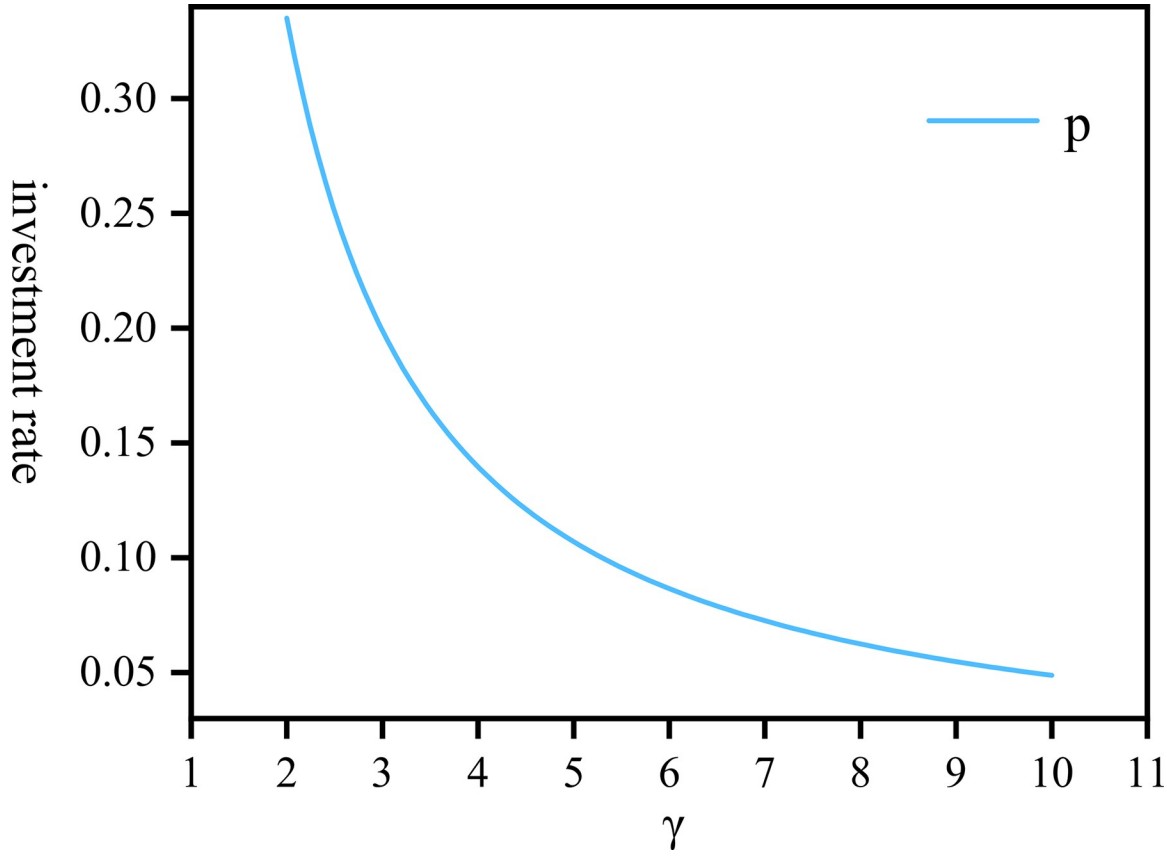

**Fig 6. The personal pension function *p(t)* draw against *γ*∈[2,10].**

Fig 8 illustrates the variations in the value functions $V_1$ and $V_2$ in response to changes in the company's risk aversion coefficient. As the risk aversion coefficient $\gamma$ increases, both value functions experience varying degrees of decrease. Initially, $V_2$ is more sensitive to $\gamma$; its rapid decline leads it to approach $V_1$. Eventually, two changes gradually converge, but this convergence does not imply equivalence between the two values. Upon closer examination of the interval [5,10], it becomes apparent to find that $V_2 > V_1$. However, they approach each other but do not become equal. This discrepancy arises because the values of $u_1^*$ and $u_2^*$ are not identical at the same $\gamma$, and there always exists a difference between them, where $u_1^* > u_2^*$.

To maximize both value functions, the company should exhibit a lower risk aversion coefficient, ideally approaching $\gamma = 2$. At this juncture, the proportion of investment in the fund $u_1^* = 0.625$, the community pension rate $u_2^* = 0.04$, the company's value function $V_1 = 31.54956$, and the employees' value function $V_2 = 283.54$ are optimized. However, in real-world scenarios, individuals' attitudes toward risk cannot be precisely quantified but are rather vaguely estimated. Hence, several recommendations are proposed for the company within the game: maintain rationality and manage investment risks prudently, carefully plan investment directions and proportions, and avoid overreacting to market disturbances or ignoring market changes' impacts on investment directions and risks. For involved employees, aligning with the company's strategies is deemed beneficial.

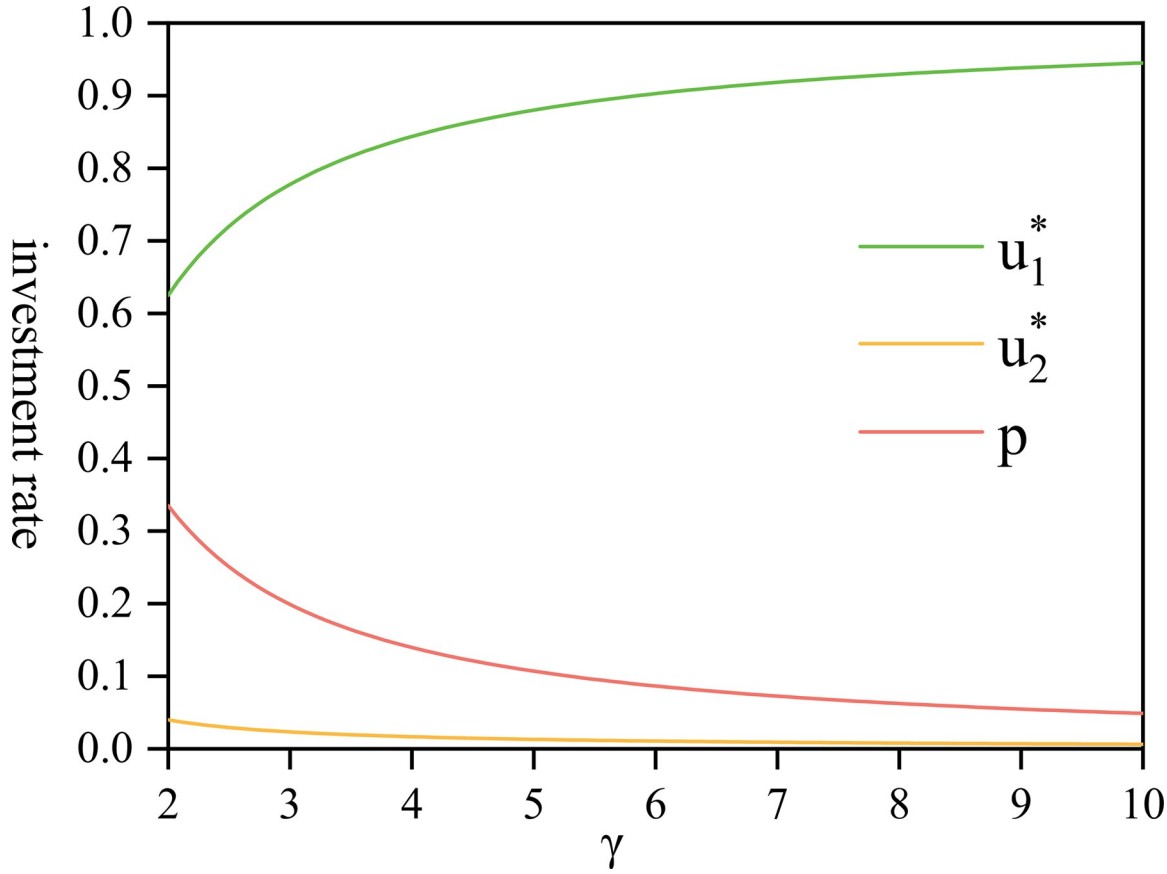

**Fig 7. The comparison of investment rate, community pension ratio, and personal pension ratio with $\gamma \in [2,10]$.**

## 5. Conclusions

Incorporating community pension into the defined benefit pension model, this study examines the dynamic interplay between the company's control over fund investment rates and employees' demand for a greater proportion of community pension. Through the establishment of a differential game framework for defined benefit pensions with community pension elements, we derive the Markov perfect Nash equilibrium solution and optimal portfolios, considering deterministic pension functions $u_1$ and the employee controlling the community pension proportion $u_2$.

Analysis reveals that under deterministic individual welfare functions, a differential game model for defined benefit pensions with community pension factors is developed and analyzed with the CRRA utility function as the objective function. Apart from that, the Markov perfect Nash equilibrium solution was used to obtain a pair of constants related to the degree of risk aversion, discount factor, risk-free rate, and Sharpe ratio. Notably, an increase in risk aversion, risk-free rate, and risk volatility results in a decrease in both the equilibrium investment proportion and the proportion invested in community pension. More importantly, as the risk aversion factor $\gamma$ increases, the investment proportion $u_1^*$ grows while the community pension proportion $u_2^*$ diminishes and becomes more sensitive to changes in $\gamma$ when $\gamma$ is small. This adjustment signifies the company's inclination to minimize investments in risk-averse assets, prioritizing risk-free assets to ensure the pension plan's sustained health and longevity. Consequently, fostering rationality in investment decisions, prudent risk management, and accurate

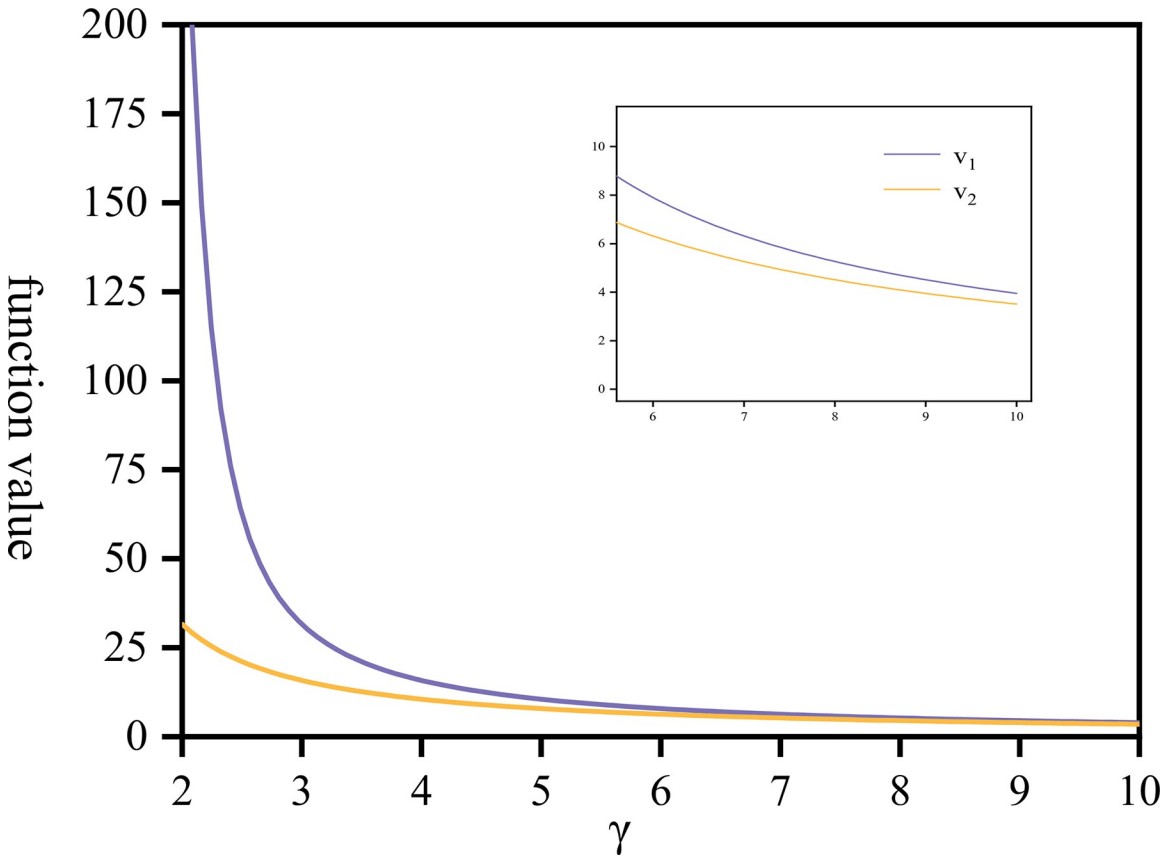

**Fig 8. The value function $V_1$, $V_2$ draw against $\gamma \in [2,10]$.**

investment planning is advised for the company. Whereas, employees are encouraged to prioritize the sustainable operation of the community pension plan over excessive capital demands.

Although this study introduces community pension into the defined benefit pension model based on China's national conditions and pension operations, the relatively simplistic objective function suggests the potential for further exploration in the integration of community pension and defined benefit pension. Future research directions could include exploring Pareto equilibrium solutions based on these findings and investigating the stochastic differential game of defined benefit pensions with community pension factors under the CEV model.

In conclusion, this study pioneers the integration of community pension into the defined benefit pension model, offering innovative insights into pension management strategies. Through the lens of a stochastic differential game, we quantitatively analyze the dynamics of defined benefit pensions under the influence of community pension factors, providing a robust framework for decision-making in pension investment. Our findings underscore the importance of considering community pensions alongside traditional pension schemes, highlighting avenues for enhancing retirement satisfaction and ensuring the long-term viability of pension plans. Further research avenues, such as exploring Pareto equilibrium solutions and investigating stochastic differential games under the CEV model, offer promising directions for advancing our understanding of pension dynamics in evolving socio-economic contexts.

A recommendation stemming from this study is to extend the analysis to encompass a broader array of pension models and socio-economic contexts to enhance the generalizability

of findings. Additionally, while this research provides valuable insights into the integration of community pension into the defined benefit pension framework, its focus on deterministic functions limits its applicability to more complex scenarios. Therefore, future studies could explore stochastic differential games under various economic conditions, considering factors like inflation, interest rate fluctuations, and market volatility to provide a more comprehensive understanding of pension dynamics. Such endeavors would enrich our understanding of pension management strategies and contribute to the development of more robust and adaptable pension systems.

## Acknowledgments

We thank the anonymous reviewers whose comments and suggestions greatly help improve and clarify this manuscript.

## Author Contributions

**Conceptualization:** Jun Wang.

**Writing – original draft:** Chunli Cui.

**Writing – review & editing:** Tian Tian.

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
