## [Decision Letter · Decision Letter 0]

12 Jan 2024

PONE-D-23-41707Equilibrium Strategy In A Defined Benefit Pension Plan With Community Pension ModelPLOS ONE

Dear Dr. Tian,

Thank you for submitting your manuscript to PLOS ONE. After careful consideration, we feel that it has merit but does not fully meet PLOS ONE’s publication criteria as it currently stands. Therefore, we invite you to submit a revised version of the manuscript that addresses the points raised during the review process.The authors need to elaborate on the advantages of the model used in this study over others.It’s better to provide a table or pictorial of existing study related to this topic.The English language of the manuscript thoroughly needs revision for better readability.Please submit your revised manuscript by Feb 26 2024 11:59PM. If you will need more time than this to complete your revisions, please reply to this message or contact the journal office at plosone@plos.org. Please include the following items when submitting your revised manuscript:A rebuttal letter that responds to each point raised by the academic editor and reviewer(s). You should upload this letter as a separate file labeled 'Response to Reviewers'.A marked-up copy of your manuscript that highlights changes made to the original version. You should upload this as a separate file labeled 'Revised Manuscript with Track Changes'.An unmarked version of your revised paper without tracked changes. You should upload this as a separate file labeled 'Manuscript'.

We look forward to receiving your revised manuscript.

Kind regards,

Sadia Ilyas, Ph.D.

Academic Editor

PLOS ONE

Journal Requirements:

4. Thank you for stating the following financial disclosure: "This work was supported by Natural Science Foundation of Jilin Province(CN) [grant number 20200201273JC] and National Natural Science Foundation of China (NSFC)[grant number 11871244]."

5. We note that your Data Availability Statement is currently as follows: "All relevant data are within the manuscript and its Supporting Information files."

Additional Editor Comments:

Reviewer-1

General comments

This paper studies the equilibrium strategy for a defined benefit pension plan with

community pension model. The authors establish a stochastic differential game model

of pension under community pension, and using the Hamilton-Jacobi-Bellman equation,

obtain the Markov perfect Nash equilibrium strategy. Finally, a numerical illustration

shows the influence of different risk aversion on the equilibrium strategy.

Overall, this paper’s language expression is poor, the logic is not particularly clear,

there are few and outdated literature used, and there are also formatting issues. In short,

this work is somewhat rough.

Specific comments

• Page 3, multiple spaces in line 57, and fewer spaces in line 64. Similar issues also exist in other pages.

• Page 3, What does the symbol “t” refer to and is there a range of it?

• Page 5, “C(t)” in equation (5) is the contribution amount, not the contribution rate.

• It is necessary to explain the meaning of the symbol γ” in Page 6 line 119 and “δ” in Page 6 line 120.

• The definition of equilibrium strategy, and the verification theorem for solutions are not mentioned in this paper.

• Proposition 3.1 in Page 7, how the statement “the constants A and B mentioned in equations (8) and (9) can be determined constant A is negative and constant B is positive“ obtain?

• Page 8, how the HJB equations obtain in lines 153-155?

• Page 10 line 191, how “0.8“ obtain?

• In Fig 1, the proportion of investment in risky assets increases with the increase of the risk aversion, this is not in line with the actual investment situation.

• The reference is scarce and outdated

Reviewer-2

PONE-D-23-41707

1. In the present manuscript, the authors analyze an equilibrium strategy for a defined benefit plan with a community pension model by employing the Hamilton-Jacobi-Bellman equation. The authors determined the Markov-perfect Nash equilibrium solution by optimizing the portfolios and yielding the risk tolerance. The study is good in terms of correlating the unfunded actuarial liability and the supplemental contribution rate; however, it needs several clarifications, which should be added before the final consideration of the manuscript.

2. The authors need to elaborate on the advantages of the model used in this study over others.

3. Additionally, it would be better if the authors could underline the limitations of this study that need further study (as suggested in the conclusion section).

4. The captions of all the figures are not defined in such a manner that the readers could get the idea by going through the figure alone. Hence, all the experimental conditions and considerations should be given in a descriptive manner.

5. The conclusion section is a weak point. The authors should highlight the findings of their study, including some numerical values.

6. The English language of the manuscript thoroughly needs revision for better readability. The authors should pay attention in this direction.

Reviewers' comments:

Reviewer's Responses to Questions

**Comments to the Author**

1. Is the manuscript technically sound, and do the data support the conclusions?

Reviewer #1: No

Reviewer #2: Yes

2. Has the statistical analysis been performed appropriately and rigorously? 

Reviewer #1: No

Reviewer #2: Yes

3. Have the authors made all data underlying the findings in their manuscript fully available?

Reviewer #1: Yes

Reviewer #2: Yes

4. Is the manuscript presented in an intelligible fashion and written in standard English?

Reviewer #1: No

Reviewer #2: No

5. Review Comments to the Author

Reviewer #1: Specific comments

• Page 3, multiple spaces in line 57, and fewer spaces in line 64. Similar issues also exist in other pages.

• Page 3, What does the symbol “t” refer to and is there a range of it?

• Page 5, “C(t)” in equation (5) is the contribution amount, not the contribution rate.

• It is necessary to explain the meaning of the symbol γ” in Page 6 line 119 and “δ” in Page 6 line 120.

• The definition of equilibrium strategy, and the verification theorem for solutions are not mentioned in this paper.

• Proposition 3.1 in Page 7, how the statement “the constants A and B mentioned in equations (8) and (9) can be determined constant A is negative and constant B is positive“ obtain?

• Page 8, how the HJB equations obtain in lines 153-155?

• Page 10 line 191, how “0.8“ obtain?

• In Fig 1, the proportion of investment in risky assets increases with the increase of the risk aversion, this is not in line with the actual investment situation.

• The reference is scarce and outdated.

Reviewer #2: PONE-D-23-41707

1. In the present manuscript, the authors analyze an equilibrium strategy for a defined benefit plan with a community pension model by employing the Hamilton-Jacobi-Bellman equation. The authors determined the Markov-perfect Nash equilibrium solution by optimizing the portfolios and yielding the risk tolerance. The study is good in terms of correlating the unfunded actuarial liability and the supplemental contribution rate; however, it needs several clarifications, which should be added before the final consideration of the manuscript.

2. The authors need to elaborate on the advantages of the model used in this study over others.

3. Additionally, it would be better if the authors could underline the limitations of this study that need further study (as suggested in the conclusion section).

4. The captions of all the figures are not defined in such a manner that the readers could get the idea by going through the figure alone. Hence, all the experimental conditions and considerations should be given in a descriptive manner.

5. The conclusion section is a weak point. The authors should highlight the findings of their study, including some numerical values.

6. The English language of the manuscript thoroughly needs revision for better readability. The authors should pay attention in this direction.

6. PLOS authors have the option to publish the peer review history of their article (what does this mean?). If published, this will include your full peer review and any attached files.

Reviewer #1: No

Reviewer #2: No

---

## [Author Response · Author response to Decision Letter 0]

27 Feb 2024

I've answered the editors' and reviewers' questions in the file, and I've uploaded the document for you to review.

---

## [Editor Report · Decision Letter 1]

5 Mar 2024

Optimal Tactics in Community Pension Model for Defined Benefit Pension Plans

PONE-D-23-41707R1

Dear Dr. Tian Tian,

We’re pleased to inform you that your manuscript has been judged scientifically suitable for publication and will be formally accepted for publication once it meets all outstanding technical requirements.

Kind regards,

Sadia Ilyas, Ph.D.

Academic Editor

PLOS ONE
---

## [Editor Report · Acceptance letter]

27 Aug 2024

PONE-D-23-41707R1 

PLOS ONE

Dear Dr. Tian, 

I'm pleased to inform you that your manuscript has been deemed suitable for publication in PLOS ONE. Congratulations! Your manuscript is now being handed over to our production team.

Kind regards, 

on behalf of

Prof. Sadia Ilyas 

Academic Editor

PLOS ONE